# Effects of Compound Probiotics on Growth Performance, Serum Biochemical and Immune Indices, Antioxidant Capacity, and Intestinal Tissue Morphology of Shaoxing Duck

**DOI:** 10.3390/ani12223219

**Published:** 2022-11-21

**Authors:** Hanxue Sun, Tiantian Gu, Guoqin Li, Li Chen, Yong Tian, Wenwu Xu, Tao Zeng, Lizhi Lu

**Affiliations:** State Key Laboratory for Managing Biotic and Chemical Threats to the Quality and Safety of Agro-Products, Institute of Animal Husbandry and Veterinary Medicine, Zhejiang Academy of Agricultural Sciences, Hangzhou 310021, China

**Keywords:** compound probiotics, duck, serum biochemical indices, intestinal tissue morphology

## Abstract

**Simple Summary:**

The Chinese duck industry volume accounts for about 80% of the world’s total. Feed is the largest duck production cost; meeting feed requirements is important for successful duck farming. As feed additives, compound probiotics play an important role in duck production. This experiment was conducted to investigate the effects of compound probiotics (CP, composed of *B. subtilis* and *B. Licheniformis* ≥ 6.0 × 10^8^ CFU/g) on growth performance, serum biochemical and immune indices, antioxidant capacity, and the intestinal tissue morphology of Shaoxing ducks. Results showed that the compound probiotics improved the growth performance, increased serum biochemical and immune indices, increased antioxidant capacity, and improved the intestinal tissue morphology of Shaoxing ducks; different day ages had different effects. This study provides a theoretical basis for the application of compound probiotics in the production of ducks and a reference basis for the development of green feed additives.

**Abstract:**

This experiment was conducted to investigate the effects of compound probiotics on growth performance, serum biochemical and immune indices, antioxidant capacity, and the intestinal tissue morphology of Shaoxing ducks. A total of 640 1-day-old healthy Shaoxing ducks of similar body weight were randomly divided into two treatment groups with eight replicates each and forty ducks per replicate. The ducks were fed a basal diet (Ctrl) or a basal diet supplemented with 0.15% compound probiotics (CP) for 125 d. The results revealed that the live body weight (BW; day 85 and 125) and the average daily gain (ADG; 28–85 and 85–125 d) of the CP group were significantly higher (*p* < 0.05) than those of the Ctrl group. In the CP group, total protein and total cholesterol contents were significantly increased (*p* < 0.05) on days 28 and 85, while triglyceride and low-density lipoprotein contents were significantly decreased (*p* < 0.05) on day 85. Furthermore, interferon-γ content was significantly increased (*p* < 0.05) in the CP group on days 28, 85, and 125. Interleukin-2 content was significantly increased (*p* < 0.05) in the CP group on days 28 and 85. Interleukin-4 content was significantly decreased (*p* < 0.05) in the CP group on day 85. Moreover, in the CP group, superoxide dismutase content was significantly increased (*p* < 0.05) on days 28 and 125, and glutathione peroxidase content was significantly increased (*p* < 0.05) on day 125. The crypt depth (CD) in the duodenum of the CP group was significantly decreased (*p* < 0.05) on days 28 and 125, whereas the villus height (VH) in the jejunum of the CP group was significantly increased (*p* < 0.05) on days 85 and 125. The VH/CD ratio in the ileum of the CP group was significantly increased (*p* < 0.05) on days 28 and 85. The VH in the ileum of the CP group was significantly increased (*p* < 0.05) on day 28. The CD in the ileum of the CP group was significantly decreased (*p* < 0.05) on day 28. In summary, the compound probiotics improved the growth performance, increased serum biochemical and immune indices, increased antioxidant capacity, and improved the intestinal tissue morphology of Shaoxing ducks.

## 1. Introduction

The Chinese duck industry volume accounts for about 80% of the world’s total. Feed is the largest duck production cost; meeting feed requirements is important for successful duck farming [1]. Shaoxing duck (*Anas platyrhynchos*), with a high egg-laying rate and a small somatotype, is an excellent egg-laying duck breed in China [2]. Improving the health of the ducks during the early stages of egg laying is particularly important for maintaining the production performance of the ducks. Research shows that probiotics can improve the intestinal and body health status of broilers by increasing intestinal barrier function, antioxidative capacity, and immunity [3]. In contrast to the effects of antibiotics, the non-specific immunity of animals is enhanced by probiotics, the utilization rate of feed is improved, the growth performance of animals is improved, they do not give rise to super-drug-resistant bacteria, and they reduce environmental pollution [4].

*Bacillus subtilis* is a Gram-positive bacteria species demonstrating probiotic properties [5]. *B. licheniformis* is widely used in a great many fields. It is a precursor component of active factors such as enzymes and antibacterial and anti-cancer substances [6,7], and it involves exopolysaccharide (EPSs) in the synthesis process [8,9,10]. Furthermore, *B. Subtilis*-supplemented diets improve growth rates, serum immunoglobulin level, intestinal microbiocenosis, and *Escherichia coli* (*E. coli*) resistance in chickens [11]. Kruger’s study found a positive effect of probiotic supplements in broiler chickens’ diet on production performance [12]. In the past decades, antibiotics, as medicine and growth promoters, have been commonly applied in animal production. Due to increased antibiotic residues and bacterial resistance, as well as the ban on using antibiotics in the feed industry in China and Europe, there is a growing interest in natural substances as alternatives to antibiotics [13,14]. Probiotics can influence the colonization of dominant intestinal microflora and reduce the damage of pathogenic bacteria to the intestinal tract [15]. Probiotics are known to be beneficial to their host and do not produce drug resistance (superbacteria). As a feed additive, they are widely considered as one of the most effective alternatives to antibiotics [16,17]. Our previous study showed that the addition of complex probiotics to the diet improved the microbiome and metabolome of the cecum contents of Shaoxing ducks [18]. Therefore, the next question was whether these improvements were partly attributable to the addition of complex probiotics to the diets, which improved antioxidant capacity and immune function in Shaoxing ducks fed antibiotic-free diets. Therefore, the aim of this study was to evaluate the beneficial effects of complex probiotics on growth performance, serum biochemical and immune indices, antioxidant capacity, and the intestinal histomorphology of Shaoxing ducks.

## 2. Materials and Methods

### 2.1. Experimental Design

A total of 640 1-day-old healthy Shaoxing ducks of similar body weight were randomly divided into two treatment groups with eight replicates each and forty ducks per replicate. The ducks were fed a basal diet (Ctrl) or a basal diet supplemented with 0.15% compound probiotics (CP) for 125 d.

### 2.2. Experimental Materials and Feeding Management

The experiment was carried out in a duck factory in Shaoxing, Zhejiang Province. The ducks, raised in a 3-layer cage duck house, were free to eat and drink daily during 16 h of light. All experiment procedures were approved by the Animal Use Committee of the Zhejiang Academy of Agricultural Sciences (2022ZAASLA40). The basal diets were formulated to meet the nutrient requirements of the egg duck standard of China (GB/T 41189-2021) combined with the physiological and nutritional requirements of ducks, as shown in Table 1. Probiotic preparation (composed of *B. subtilis* and *B. Licheniformis*; ≥6.0 × 10^8^ CFU/g) was provided by Guangzhou Hengyi Biotechnology Co., Ltd. (Guangzhou, China).

### 2.3. Sample Collection

Body weight (BW) was determined (Electronic bench scale, Beijing Zhebei Technology Co., Ltd. 6541Q, Beijing, China) on days 28, 85, and 125 before feeding, and the average daily gain (ADG) was calculated. Additionally, on days 28, 85, and 125, blood was collected with disposable vacuum blood collection tubes from the wing vein of each duck. The blood samples (5 mL) were centrifuged at 3000 rpm for 15 min at 4 °C to collect serum and then stored at −20 °C until further analysis. To determine intestinal morphology after the completion of the trial, 8 healthy ducks were randomly selected, and the duodenum, jejunum, and ileum samples were obtained after humanely sacrificing and dissecting the ducks.

#### 2.3.1. Serum Index Determination

Serum total protein (TP), total cholesterol (TC), triglyceride (TG), high-density lipoprotein (HDL), low-density lipoprotein (LDL), and γ-glutamyl transpeptidase (γ-GT) were analyzed using automatic biochemical analyzer (HATICHI 7180, Tokyo, Japan). The immunoglobulin M (IgM), immunoglobulin G (IgG), immunoglobulin A (IgA), interferon-γ (IFN-γ), interleukin-2 (IL-2), interleukin-4 (IL-4), tumor necrosis factor-α (TNF-α), superoxide dismutase (SOD), total antioxidant capacity (T-AOC), catalase (CAT), malondialdehyde (MDA), and glutathione peroxidase (GSH-Px) were determined with the multifunctional marker (SuPerMax 3100, Shanghai, China). Enzyme-linked immunoassay kits were purchased from Shanghai Sangon Biotechnology Co., Ltd. (Shanghai, China). The detection assays were conducted as per the manufacturer’s instructions.

#### 2.3.2. Intestinal Morphology

Each intestinal segment was fixed in 4% paraformaldehyde for more than 24 h (EG1150h, LEIC, Wetzlar, Germany). The tissue was taken out of the fixative and a scalpel used to smooth the target tissue in the fume hood. The cut tissue and the corresponding label were put in the dehydration box. The dehydration box was put into the basket, and the alcohol was dehydrated in order in the dehydrator. The wax-soaked tissues were embedded in the embedding machine (rotating microbodies, RM2225, LEIC, Wetzlar, Germany). The embedded wax block was fixed on the slicer and cut into thin slices, generally 5–8 microns thick. They were dried in a 45 °C constant-temperature oven. The paraffin was removed from the slice and stained with hematoxylin and eosin [19]. Microscopic examination for the villus height (VH) and crypt depth (CD) was carried out under a light microscope (S4E, LEIC, Wetzlar, Germany) and image analyzer (Image-Proplus 5.0).

### 2.4. Statistical Analysis

Statistical analyses were conducted using SPSS (SPSS version 22.0; IBM Corp, Armonk, NY, USA) statistics software. Data are expressed as mean ± SEM. A *p*-value < 0.05 in the Student’s *t*-test was considered to be statistically significant.

## 3. Results

### 3.1. Effects of Compound Probiotics on Growth Performance of Ducks

The BW (days 85 and 125) and ADG (28–85 d, 85–125 d, and 1–125 d) of the CP group were significantly higher (*p* < 0.05) than those of the Ctrl group (Table 2). Furthermore, the results revealed that the initial BW (day 28) and ADG (days 1–28) were not significantly affected by the dietary treatments (*p* > 0.05).

### 3.2. Effect of Compound Probiotics on Serum Biochemical Indices of Ducks

At day 28, the TP and TC contents in the CP group were significantly higher (*p* < 0.05) than those in the Ctrl group (Table 3). This trend persisted until the ducklings were 85 days old, after which the TP content in the experimental ducks was 28.94 g/L, which was higher than that in the control ducklings (*p* < 0.01). The TC and TG contents in the CP group were significantly lower (*p* < 0.01) than those in the Ctrl group. The LDL content in the CP group was significantly lower (*p* < 0.05) than that in the Ctrl group. At day 125, the TG content in the CP group was significantly lower (*p* < 0.01) than that in the Ctrl group.

### 3.3. Effect of Compound Probiotics on Serum Immune Indices of Ducks

At days 28 and 85, the IFN-γ and IL-2 contents in the CP group were significantly higher (*p* < 0.05) than those in the Ctrl group (Table 4). At day 85, the IL-4 content in the CP group was significantly lower (*p* < 0.05) than that in the Ctrl group. At day 125, the INF-γ content in the CP group was significantly higher (*p* < 0.05) than that in the Ctrl group.

### 3.4. Effect of Compound Probiotics on Serum Antioxidative Indices of Ducks

At days 28 and 125, the SOD content in the CP group was significantly higher (*p* < 0.05) than that in the Ctrl group (Table 5). Additionally, at day 125, the GSH-Px content in the CP group was significantly higher (*p* < 0.05) than that in the Ctrl group.

### 3.5. Effect of Compound Probiotics on the Morphological Structure in the Duodenum Mucosa of Ducks

At days 28 and 125, the CD in the duodenum of the CP group was significantly lower (*p* < 0.05) than that of the Ctrl group (Table 6, Figure 1). However, there was no significant difference (*p* > 0.05) in the VH/CD ratio between the CP and the Ctrl groups.

### 3.6. Effect of Compound Probiotics on the Morphological Structure in the Jejunum Mucosa of Ducks

At days 85 and 125, the VH in the jejunum of the CP group was significantly higher (*p* < 0.05) than those of the Ctrl group (Table 7, Figure 2). However, there was no significant difference (*p* > 0.05) in the CD and the VH/CD ratio between the CP and the Ctrl groups.

### 3.7. Effect of Compound Probiotics on the Morphological Structure in the Ileum Mucosa of Ducks

At day 28, the VH and VH/CD in the ileum of the CP group were significantly higher (*p* < 0.05) than those of the Ctrl group, while the CD in the ileum of the CP group was significantly lower (*p* < 0.05) than that of the Ctrl group (Table 8, Figure 3). At day 85, the VH/CD in the ileum of the CP group was higher (*p* > 0.05) than that in the Ctrl group.

## 4. Discussion

Several studies reported that probiotics are beneficial to growth performance [20]. Aliakbarpour et al. [21] reported that body weight at 42 days of age was higher in broiler chickens fed a diet with probiotics compared to that of those fed a diet without probiotics. Moreover, diets supplemented with 0.05% probiotics (Livesac, Zeus Biotech Limited, India) significantly increased (*p* < 0.01) the live weight of Peking ducks (8 weeks old) [22]. Feed is the largest duck production cost; meeting feed requirements is important for successful duck farming. One study revealed that an increase in the digestibility of feed components and an improvement in feed conversion are the primary effects of probiotic supplements [23]. In this experiment, the live BW (days 85 and 125) and the ADG (28–85 d, 85–125 d, and 1–125 d) of the CP group were significantly higher than those of the Ctrl group. Probiotics may improve the feed conversion efficiency of Shaoxing ducks; however, the initial BW (day 28) and ADG (1–28 d) were not significantly affected by the dietary treatments. This could be because the duration was too short for the probiotics to fully play their role.

Feed supplemented with *Lactobacilli* and *Bifidobacteria* stimulated the protein metabolism [23]. Additionally, the TP and albumin levels in the blood serum of 42-day-old ducklings fed probiotic-supplemented feed increased significantly. Another study [24] also demonstrated that *Saccharomyces* cerevisiae-supplemented feed resulted in a significant increase in TP levels in the serum of broiler ducklings. In this experiment, at 28 days of age, the serum TP and TC contents of the CP group were higher than those of the Ctrl group. This trend persisted until the ducklings were 85 days old, after which the TP content in the experimental ducks was 28.94 g/L, which was higher than that in the ducks of the Ctrl group. At days 85 and 125, the serum TC and TG contents of the CP group were lower than those of the Ctrl group. Therefore, the results reveal that *B. subtilis*- and *B. licheniformis*-supplemented diets can reduce the serum TG content and improve the health of Shaoxing ducks.

Immunoglobulins are a globulin protein produced by animals in response to antigens. They bind to antigens and activate complement proteins. Immunoglobulin G (IgG) content is the highest in the serum, and it has antibacterial and antiviral properties and contributes important immunological activities, such as the regulation, agglutination, and precipitation of antigens [25]. IL-2 can promote T cell growth and regulate humoral immune response [26]. In this study, at days 28 and 85, the serum INF-γ and IL-2 contents of the CP group were higher than those of the Ctrl group. At day 85, the serum IL-4 content of the CP group was significantly lower than that of the Ctrl group. At day 125, the serum INF-γ content of the CP group was significantly higher than that of the Ctrl group. A study by Guo et al. [11] revealed that diets supplemented with *B. subtilis* could trigger toll-like receptor 4 to enhance the immune ability of the body in chickens.

The antioxidant system can be divided into non-enzymatic and enzymatic systems; the antioxidant enzymes in the enzymatic system include SOD, GSH-Px, and CAT [27]. The oxidation of the body can accumulate MDA, which can damage animal cells. The overall antioxidant capacity can be determined by detecting SOD, GSH-Px, CAT, and MDA contents [28]. A study by Zhao et al. [29] revealed that diets supplemented with probiotics could considerably enhance SOD and GSH-Px activities and reduce MDA level. *Bacillus* probiotics have the efficacy to significantly decrease the serum MDA content and increase the SOD [30] and GSH contents. In our study, at days 85 and 125, the serum SOD contents of the CP group were higher than those of the Ctrl group. At day 125, the serum GSH-Px content of the CP group was higher than that of the Ctrl group. *Lactobacillus* content is positively correlated with SOD activity [30]; thus, the addition of probiotics in the diet may affect the abundance of intestinal flora and increase the antioxidant capacity in the body.

In our study, the TG and TC of Shaoxing ducks were reduced by dietary supplementation with complex probiotics in the later stages of the trial. The decrease in TG and TC may be related to the inhibition of lipid peroxidation by improving the serum antioxidant capacity of Shaoxing ducks with probiotic complexes. TP, composed of albumin and globulin, reflects an organism’s protein absorption and the relationship with humoral immunity. In our study, the TP of Shaoxing ducks was improved by dietary supplementation with complex probiotics, further indicating improved immune capacity. Blood biochemical indexes can reflect the absorption and utilization rate of nutrients to a certain extent [31]. The supplementation of probiotic complexes in the diet may improve the growth performance of Shaoxing ducks by regulating the activity of serum biochemical enzymes and the level of immunologically active substances.

Improved digestion of food and absorption of nutrients in animals fed with probiotics might cause colonization of dominant intestinal microflora and reduce the damage of pathogenic bacteria to the intestinal tract [11,32]. Intestinal tissue morphology also changes under the action of probiotics, including the increase of VH and VH/CD ratio, indicating that the absorption surface area is increased, and the digestion and absorption capacities are improved [33,34]. In our study, at days 85 and 125, the CD in the duodenum of the CP group was lower than that of the Ctrl group. At days 85 and 125, the VH in the jejunum of the CP group was higher than that of the Ctrl group. At day 28, the VH and VH/CD ratio in the ileum of the CP group were higher than those of the Ctrl group. The CD in the ileum of the CP group was lower than that of the Ctrl group. Therefore, our results demonstrate that ducks fed with compound probiotics had greater VH, shallower CD, and a higher VH/CD ratio in the jejunum, which might explain the elevated growth performance of the treatment ducks.

## 5. Conclusions

The effects of compound probiotics on growth performance, serum biochemical and immune indices, antioxidant capacity, and the intestinal tissue morphology of Shaoxing ducks were assessed and measurements compared with those of the ducks in the control group. Our results demonstrated that compound probiotics can improve the growth performance, increase serum biochemical and immune indices, increase antioxidant capacity, and improve the intestinal health of Shaoxing ducks.

## Figures and Tables

**Figure 1 animals-12-03219-f001:**
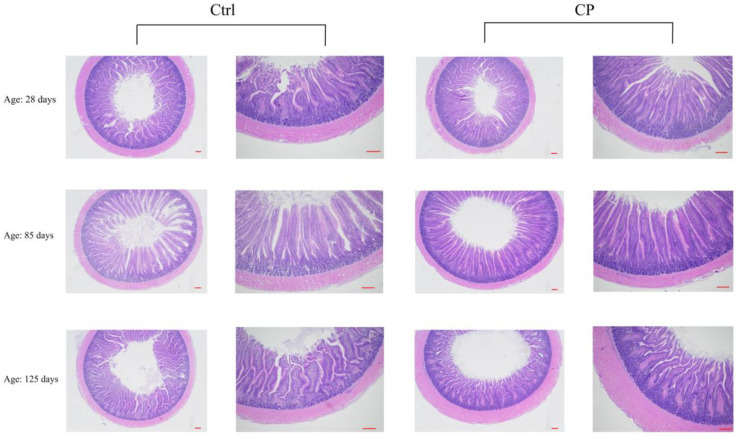
Effects of compound probiotics on the morphological structure in the duodenum mucosa of ducks. Ctrl, basal diet, CP, basal diet supplemented with 0.15% compound probiotics. The first and third columns of the pictures are scaled to 100 µm, and the second and fourth columns of the pictures are scaled to 50 µm.

**Figure 2 animals-12-03219-f002:**
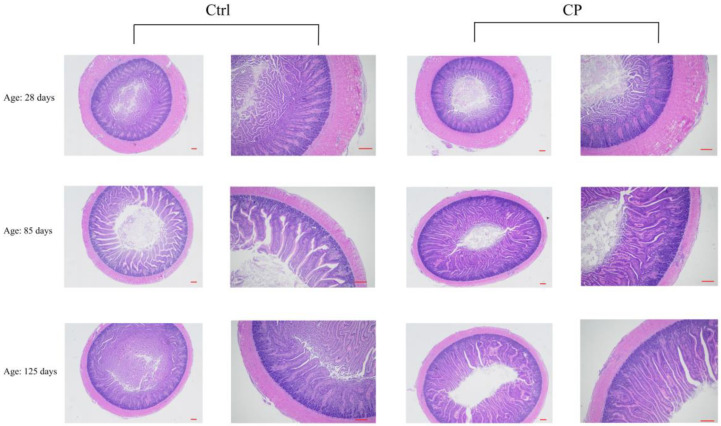
Compound probiotic effect on the morphological structure in the jejunum mucosa of ducks. Ctrl, basal diet, CP, basal diet supplemented with 0.15% compound probiotics. The first and third columns of the pictures are scaled to 100 µm, and the second and fourth columns of the pictures are scaled to 50 µm.

**Figure 3 animals-12-03219-f003:**
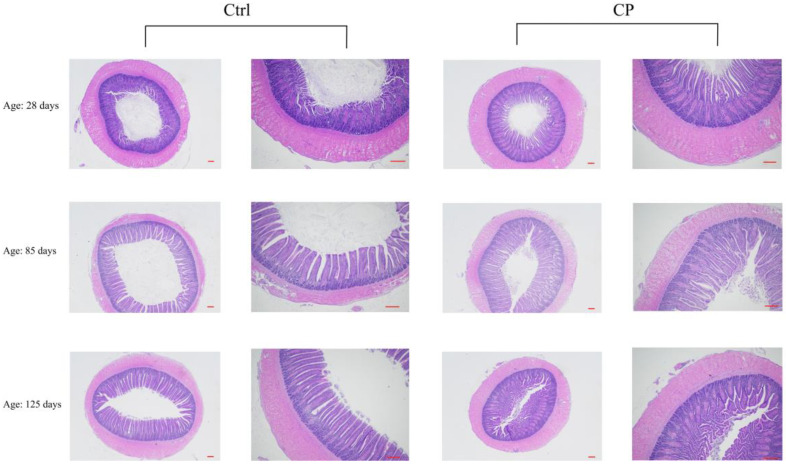
Effects of compound probiotics on the morphological structure in the ileum mucosa of ducks. Ctrl, basal diet, CP, basal diet supplemented with 0.15% compound probiotics. The first and third columns of the pictures are scaled to 100 µm, and the second and fourth columns of the pictures are scaled to 50 µm.

**Table 1 animals-12-03219-t001:** Composition and nutrient levels of the basal diet (dry; %).

Ingredients	1–28 d	85–125 d	Nutrient Levels ^b^	1–28 d	85–125 d
Corn	56.64	58.05	ME (MJ/kg)	11.37	10.97
Soybean meal	29.70	26.54	CP	19.31	16.14
Wheat bran	5.10	7.00	Lys	0.98	0.77
CaHPO_4_	1.37	1.31	Met	0.43	0.31
L-Lys	0.49	0.49	Met+Cys	0.73	0.61
DL-Met	0.20	0.16	Ca	0.94	0.81
Nacl	0.30	0.30	TP	0.65	0.40
Limestone	3.50	3.45			
Soybean oil	1.70	1.70			
Premix ^a^	1.00	1.00			
Total	100.00	100.00			

^a^ The premixed feed was provided with Cu (12 mg), Zn (84 mg), Mn (110 mg), I (10.4 mg), Se (0.3 mg), VA (11250 IU), VD3 (3000 IU), VE (37.5 mg), VK (3 mg), VB1 (4 mg), VB2 (7.2 mg), niacin (60 mg), calcium ubiquitin (14 mg), VB6 (4 mg), VB11 (1.3 mg), VB12 (0.02 mg), and choline (0.1 0 mg); measurements are per kg of diet. ^b^ Nutrient levels were calculated using NRC (1994) values.

**Table 2 animals-12-03219-t002:** Effects of compound probiotics on growth performance of ducks.

Items ^a^	Ctrl	CP	*p-*Value ^b^
Initial BW g	41.03 ± 0.44	40.28 ± 0.46	0.262
28 d BW g	649.55 ± 15.6	677.81 ± 11.72	0.137
ADG (1–28 d) g/d	21.73 ± 0.54	22.88 ± 0.40	0.118
85 d BW g	1167.71 ± 36.3	1318.58 ± 28.33	<0.05
ADG (28–85 d) g/d	18.51 ± 0.83	22.78 ± 0.62	<0.05
125 d BW g	1399.23 ± 28.16	1677.69 ± 33.05	<0.01
ADG (85–125 d) g/d	8.27 ± 0.69	12.83 ± 0.29	<0.01
ADG (1–125 d) g/d	10.86 ± 0.80	13.09 ± 0.94	<0.01

^a^ BW, body weight; ADG, average daily gain; Ctrl, basal diet; CP, basal diet supplemented with 0.15% compound probiotics; ^b^
*p*-value obtained by *t*-test.

**Table 3 animals-12-03219-t003:** Effects of compound probiotics on serum biochemical indices of ducks.

Items ^a^	Ctrl	CP	*p-*Value ^b^
Age: 28 days			
TP ((g/L)	19.57 ± 1.18	24.84 ± 1.21	<0.01
TC (mmol/L)	2.24 ± 0.26	4.34 ± 0.48	<0.05
TG (mmol/L)	0.93 ± 0.32	0.72 ± 0.21	0.596
HDL (mmol/L)	2.38 ± 0.33	3.48 ± 0.39	0.053
LDL (mmol/L)	1.73 ± 0.18	2.06 ± 0.3	0.373
γ-GT (U/L)	3.68 ± 0.4	3.76 ± 0.4	0.071
Age: 85 days			
TP (g/L)	21.33 ± 1.15	28.94 ± 1.41	<0.01
TC (mmol/L)	6.69 ± 0.70	3.49 ± 0.66	<0.01
TG (mmol/L)	3.53 ± 0.46	0.87 ± 0.11	<0.01
HDL (mmol/L)	4.43 ± 0.50	3.69 ± 0.43	0.284
LDL (mmol/L)	2.27 ± 0.30	1.46 ± 0.18	<0.05
γ-GT (U/L)	3.90 ± 0.30	3.17 ± 0.29	0.108
Age: 125 days			
TP ((g/L)	17.99 ± 1.05	15.7 ± 0.91	0.124
TC (mmol/L)	2.25 ± 0.32	1.90 ± 0.22	0.402
TG (mmol/L)	4.62 ± 0.68	0.90 ± 0.26	<0.01
HDL (mmol/L)	1.91 ± 0.55	2.18 ± 0.23	0.660
LDL (mmol/L)	0.95 ± 0.18	1.17 ± 0.15	0.377
γ-GT (U/L)	2.61 ± 0.22	2.57 ± 0.16	0.892

^a^ TP, total protein; TC, total cholesterol; TG, triglyceride; HDL, high-density lipoprotein; LDL, low-density lipoprotein; γ-GT, γ-glutamyl transpeptidase; Ctrl, basal diet; CP, basal diet supplemented with 0.15% compound probiotics; ^b^
*p*-value obtained by *t*-test.

**Table 4 animals-12-03219-t004:** Effects of compound probiotics on serum immune indices of ducks.

Items ^a^	Ctrl	CP	*p*-Value ^b^
Age: 28 days			
IgM ((g/L)	1.44 ± 0.2	1.25 ± 0.12	0.429
IgG (g/L)	3.57 ± 0.15	3.47 ± 0.33	0.775
IgA (g/L)	1.66 ± 0.22	1.28 ± 0.3	0.326
IFN-γ (pg/mL)	45.4 ± 2.04	54.55 ± 3.64	<0.05
IL-2 (pg/mL)	323.32 ± 13.44	356.47 ± 13.77	<0.05
IL-4 (pg/mL)	6.25 ± 0.41	5.47 ± 0.75	0.375
TNFα (pg/mL)	63.95 ± 3.19	71.36 ± 2.33	0.085
Age: 85 days			
IgM (g/L)	1.89 ± 0.11	1.98 ± 0.21	0.134
IgG (g/L)	4.26 ± 0.33	3.20 ± 0.37	0.053
IgA (g/L)	2.43 ± 0.15	2.27 ± 0.09	0.369
IFN-γ (pg/mL)	45.12 ± 1.12	49.99 ± 1.88	<0.05
IL-2 (pg/mL)	278.1 ± 11.68	302.44 ± 16.85	<0.05
IL-4 (pg/mL)	10.54 ± 0.32	8.22 ± 0.41	<0.05
TNFα (pg/mL)	56.58 ± 3.87	64.37 ± 3.28	0.151
Age: 125 days			
IgM (g/L)	2.26 ± 0.19	2.35 ± 0.25	0.321
IgG (g/L)	5.19 ± 0.29	5.31 ± 0.43	0.763
IgA (g/L)	3.19 ± 0.25	2.47 ± 0.13	0.412
IFN-γ (pg/mL)	34.2 ± 0.86	38.22 ± 1.12	<0.05
IL-2 (pg/mL)	234.93 ± 10.18	266.26 ± 12.52	0.076
IL-4 (pg/mL)	12.07 ± 0.6	10.38 ± 1.17	0.223
TNFα (pg/mL)	43.02 ± 1.7	51.17 ± 4.18	0.096

^a^ IgM, immunoglobulin M; IgG, immunoglobulin G; IgA, immunoglobulin A; IFN- γ, interferon-γ; IL-2, interleukin-2; IL-4, interleukin-4; TNF-α, tumor necrosis factor-α; Ctrl, basal diet; CP, basal diet supplemented with 0.15% compound probiotics; ^b^
*p*-value obtained by *t*-test.

**Table 5 animals-12-03219-t005:** Effects of compound probiotics on serum antioxidative indices of ducks.

Items ^a^	Ctrl	CP	*p-*Value ^b^
Age: 28 days			
SOD (U/mL)	43.37 ± 2.49	53.04 ± 2.72	<0.05
T-AOC (U/mL)	9.51 ± 0.21	9.29 ± 0.23	0.503
CAT (U/mL)	64.02 ± 2.47	63.55 ± 1.73	0.879
MDA (nmol/mL)	4.02 ± 0.36	4.00 ± 0.22	0.955
GSH-Px (U/mL)	310.65 ± 8.84	310.84 ± 17.44	0.992
Age: 85 days			
SOD (U/mL)	49.34 ± 1.7	49.89 ± 2.34	0.854
T-AOC (U/mL)	9.56 ± 0.13	9.40 ± 0.21	0.519
CAT (U/mL)	74.16 ± 2.42	75.48 ± 1.19	0.634
MDA (nmol/mL)	3.44 ± 0.66	3.96 ± 0.43	0.521
GSH-Px (U/mL)	337.37 ± 14.99	338.75 ± 11.44	0.292
Age: 125 days			
SOD (U/mL)	99.69 ± 2.78	105.22 ± 1.86	<0.05
T-AOC (U/mL)	10.06 ± 0.20	10.06 ± 0.33	0.995
CAT (U/mL)	77.54 ± 1.53	77.56 ± 2.99	0.964
MDA (nmol/mL)	2.41 ± 0.26	2.32 ± 0.40	0.862
GSH-Px (U/mL)	380.44 ± 8.84	395.69 ± 10.71	<0.05

^a^ SOD, superoxide dismutase; T-AOC, total antioxidant capacity; CAT, catalase; MDA, malondialdehyde; GSH-PX, glutathione peroxidase; Ctrl, basal diet; and CP, basal diet supplemented with 0.15% compound probiotics; ^b^
*p*-value obtained by *t*-test.

**Table 6 animals-12-03219-t006:** Effects of compound probiotics on the morphological structure in the duodenum mucosa of ducks.

Items ^a^	Ctrl	CP	*p*-Value ^b^
Age: 28 days			
Villi height/μm	793.91 ± 10.03	750.9 ± 21.44	0.128
Crypt depth/μm	166.52 ± 3.85	160.77 ± 3.72	<0.05
VH/CD	4.79 ± 0.15	4.68 ± 0.17	0.659
Age: 85 days			
Villi height/μm	885.32 ± 25.54	1020.61 ± 18.66	0.350
Crypt depth/μm	161.1 ± 2.18	161.51 ± 4.47	0.942
VH/CD	5.5 ± 0.15	6.36 ± 0.25	0.953
Age: 125 days			
Villi height/μm	959.91 ± 14.16	978.31 ± 33.55	0.678
Crypt depth/μm	127.06 ± 3.09	126.79 ± 2.51	<0.05
VH/CD	7.59 ± 0.24	7.73 ± 0.3	0.741

^a^ VH, villi height; CD, crypt depth; VH/CD, villi height/crypt depth; Ctrl, basal diet; CP, basal diet supplemented with 0.15% compound probiotics; ^b^
*p*-value obtained by *t*-test.

**Table 7 animals-12-03219-t007:** Effects of compound probiotics on the morphological structure in the jejunum mucosa of ducks.

Items ^a^	Ctrl	CP	*p*-Value ^b^
Age: 28 days			
Villi height/μm	694.06 ± 9.93	719.9 ± 12.59	0.601
Crypt depth/μm	159.01 ± 4.21	150.76 ± 4.55	0.235
VH/CD	4.38 ± 0.13	4.79 ± 0.30	0.282
Age: 85 days			
Villi height/μm	829.27 ± 12.44	906.44 ± 9.30	<0.05
Crypt depth/μm	151.97 ± 2.07	159.36 ± 7.36	0.394
VH/CD	5.46 ± 0.08	5.76 ± 0.25	0.330
Age: 125 days			
Villi height/μm	1066.48 ± 13.58	1122.11 ± 12.98	<0.05
Crypt depth/μm	177.68 ± 2.63	171.11 ± 6.04	0.384
VH/CD	6.01 ± 0.11	6.62 ± 0.3	0.114

^a^ VH, villi height; CD, crypt depth; VH/CD, villi height/crypt depth; Ctrl, basal diet; CP, basal diet supplemented with 0.15% compound probiotics; ^b^
*p*-value obtained by *t*-test.

**Table 8 animals-12-03219-t008:** Effects of compound probiotics on the morphological structure in the ileum mucosa of ducks.

Items ^a^	Ctrl	CP	*p*-Value ^b^
Age: 28 days			
Villi height/μm	418.70 ± 15.6	472.70 ± 7.90	<0.05
Crypt depth/μm	150.9 ± 2.48	142.96 ± 2.47	<0.05
VH/CD	2.78 ± 0.09	3.32 ± 0.11	<0.05
Age: 85 days			
Villi height/μm	492.05 ± 14.5	525.83 ± 15.08	0.171
Crypt depth/μm	155.58 ± 2.46	157.36 ± 2.55	0.656
VH/CD	3.17 ± 0.09	3.56 ± 0.08	<0.05
Age: 125 days			
Villi height/μm	664.95 ± 11.41	646.51 ± 12.37	0.340
Crypt depth/μm	121.32 ± 2.95	112.07 ± 3.92	0.116
VH/CD	5.50 ± 0.16	5.65 ± 0.21	0.599

^a^ VH, villi height; CD, crypt depth; VH/CD, villi height/crypt depth; Ctrl, basal diet; CP, basal diet supplemented with 0.15% compound probiotics; ^b^
*p*-value obtained by *t*-test.

## Data Availability

The data presented in this study are available on request from the authors.

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
