# Peer review of "Effects of Compound Probiotics on Growth Performance, Serum Biochemical and Immune Indices, Antioxidant Capacity, and Intestinal Tissue Morphology of Shaoxing Duck"

_animals, 2022, doi:10.3390/ani12223219_

Round 1
Reviewer 1 Report
This study evaluated the effects of compound probiotics on growth performance, serum biochemical and immune indices, antioxidant capacity, and intestinal tissue morphology of Shaoxing duck. This work is the first composed of B. subtilis and B. Licheniformis for the different day ages ducks and provides new knowledge of the growth performance, serum biochemical and immune indices, antioxidant capacity, and intestinal tissue morphology. Although the argument was well defined, there are some inconsistences related to rigor and quality of writing.
Minor comments:
1. Line 15-16: Whether it is necessary to indicate the composition and concentration of compound probiotics, as line 82 “Probiotic preparation (composed of B. subtilis and B. Licheniformis; ≥6.0 × 108 CFU/g)”.
2. Line 57: Should be “Gram-positive bacteria”.
3. Introduction it is recommended to add more literature studies to clarify the background.
4. Line 77: Please use case correctly.
5. Line 87-88: Number should be subscript.
6. Line 91-93: “Body weight (BW) was measured on days 28, 85, and 125 before feeding, and the average daily gain (ADG) was calculated. Additionally, on days 28, 85, and 125, blood was collected with disposable vacuum blood collection tubes from the wing vein of each duck.” Why to collect samples in these three periods.
7. Line 213-218: The author should focus on how probiotics can improve the growth performance of Shaoxing duck.
8. Line 253: Should be analyzed together with the results of serum biochemical and immune indices, antioxidant capacity.
9. Grammar and Punctuation must be reviewed.
10. Latin names should be used in italic font.
11. The first three sentences in the Abstract are not clear enough.
Reviewer 2 Report
This study demonstrated that the probiotics complex (Bacillus subtilis and Bacillus licheniformis) could improve growth performance, serum biochemical and immune indices, antioxidant capacity, and tissue morphological indexes. However, there were no connections between every result. please add some paragraphs describing the interrelationships between results. Additionally, the author said that the duck was randomly selected to conduct experimental allotment on day 0, which will be the incorrect way in an animal experiment. Please add the exact reason for choosing the above method.
Reviewer 3 Report
The manuscript ANIMALS-1955225 entitled ' Effects of Compound Probiotics on Growth Performance, Serum Biochemical and Immune Indices, Antioxidant Capacity, and Intestinal Tissue Morphology of Shaoxing Duck' falls with its subject within the scope of ANIMALS. This study provides a theoretical basis for the application of the compound probiotics application in the production of ducks. However, there are a number of small problems in the paper which the authors should easily be able to rectify to improve it.
Check the manuscript thoroughly for the English language and correct it to improve the grammar, style and readability.
L21: in the production of squabs? Please check.
L26-27: Authors use both “Control” and “Ctrl” as the group name which ducks with basal diets. This is not to be recommended.
L123: add the detailed information of “SPSS”
L127/Table2: Please add the data of growth performance in the whole period. Also, I suggest that the authors follow the standard method of presenting results in the Results section.
Fig. 1/2: Please add the scale bar of the pictures.
Reviewer 4 Report
The purpose of this study was to determine the eefects of compound probiotics on growth performance, serum biochemical and immune indices, antioxidant capacitym and intestinal tissue morphometry of Shaoxing ducks. The Introduction chapter provides an overview of the world's knowledge on this subject. The material used in the research is sufficiently numerous, but some supplementing the description in Materials and Methods chapter are needed. The results are described usually correctly. The discussion is exhaustive. Summary of the results are correct. Some corrections are needed before publishing an article in Animals. The proposed changes are listed below.
General comments:
Please prepare the article in accordance with the instructions for authors:
For significance please use lowercase "p" in italic instead of uppercase "P" throughout the main article
In the Author Contributions section, use the one activity, then list the authors, the second activity, then the performers (authors), see other published articles in Animals
Detailed comments:
L13 „a dot” instead of a comma after "total"
L21 squabs - a term used rather for slaughter pigeons, not for young slaughter ducks,
L31 - LDL content was slightly higher at 125 days
L33 no description of IL-2 (on days 28 and 85) and IL-4 (on days 85)
L35 SOD content was significantly increased on days 28 and 125, but not on days 85
L40 no description for VH for 28 days
L46 dot after "total", not a comma
L48 space before (Anas ... "
L59 space before [6,7]
L60 space before [8-10]
L73 + provide information on building type (closed, no windows?), floor type, density/m2; temperature, humidity, lighting program (length, intensity, color)
L91 measured BW or determined BW?
L91 add scale name, producer data, quantity of measurement who used for dteremined BW
L96 identical growt - what does it mean?
L112 space before (EG1150h ... "
L119 space before [16]
L124,129,134 "p in italics, with lowercase letter
L137 space before (Table 3)
L160 - 28 and 125 not 85 and 125, SOD content in the CP group
L170 28 and 125, not 85 and 125
L183 "p" - italics, lowercase
L184 in the evaluation deadlines?
L256 [11,29], no spaces
L305,350 remove "dot" after (Basel)
L310 removes "dot" after "Microorganisms"
L341 "Res." or Research?
Round 2
Reviewer 2 Report
The added paragraph in Discussion session is very appropriate. However, the curiosity about the method of allotment still remains. Generally, when an animal experiment begins, the allotment must be performed according to the animal's body weight to eliminate the variation from the animal's growth rate. In this experiment, the initial body weight has a p-value of 0.262, and the authors would be considered the initial body weight between the control and treatment to be the same. However, in aspects of the intestinal development state, it seems that there is a possibility that the gut environment is different. Collectively, this experiment would be difficult to be justified without verification of the random selection of ducklings on Day 1.
